# Ecosystem structural changes controlled by altered rainfall climatology in tropical savannas

Wenmin Zhang[1,2], Martin Brandt [1], Josep Penuelas [3,4], Françoise Guichard[5], Xiaoye Tong[1], Feng Tian [1,6] & Rasmus Fensholt [1]

Tropical savannas comprise mixed woodland grassland ecosystems in which trees and grasses compete for water resources thereby maintaining the spatial structuring of this ecosystem. A global change in rainfall climatology may impact the structure of tropical savanna ecosystems by favouring woody plants, relative to herbaceous vegetation. Here we analysed satellite data and observed a relatively higher increase in woody vegetation (5%) as compared to the increase in annual maximum leaf area index ($LAI_{max}$, an indicator of the total green vegetation production) (3%) in arid and semi-arid savannas over recent decades. We further observed a declining sensitivity of $LAI_{max}$ to annual rainfall over 56% of the tropical savannas, spatially overlapping with areas of increased woody cover and altered rainfall climatology. This suggests a climate-induced shift in the coexistence of woody and herbaceous vegetation in savanna ecosystems, possibly caused by altered hydrological conditions with significance for land cover and associated biophysical effects such as surface albedo and evapotranspiration.

[1] Department of Geosciences and Natural Resource Management, University of Copenhagen, 1350 Copenhagen, Denmark. [2] International Institute for Earth System Sciences, Nanjing University, 210023 Nanjing, China. [3] CSIC, Global Ecology Unit CREAF-CSIC-UAB, Bellaterra, 08193 Catalonia, Spain. [4] CREAF, Cerdanyola del Vallès, 08193 Catalonia, Spain. [5] Centre National de Recherches Météorologiques (CNRM), UMR CNRS 3589 and Météo-France, 42 Avenue Gaspard Coriolis, 31057 Toulouse, France. [6] Department of Physical Geography and Ecosystem Science, Lund University, 22362 Lund, Sweden. Correspondence and requests for materials should be addressed to W.Z. (email: wenminzhg@gmail.com)

Tropical savanna ecosystems are characterized by the coexistence of woody and herbaceous vegetation[1,2], the distribution and abundance of which are controlled by herbivores, fire and rainfall[3,4]. Changing patterns in one of these controlling factors can lead to an imbalance of this coexistence, ultimately causing changes in the structure and functioning of savanna ecosystems. It has been well documented that increasing pressure by herbivores (in particular livestock) favours woody plants at the detriment of herbaceous vegetation, causing a phenomenon termed woody encroachment, which is widespread across global savannas[5] and in particular in Sub-Saharan Africa[6]. Moreover, a declining fire frequency has been observed[7], also known to sustain the growth of woody vegetation in savanna ecosystems[5]. Recently, increased atmospheric $CO_2$ concentrations were suggested to increase the growth rate of trees relative to grasses[5,8–10]. The impact from changes in rainfall on the vegetation production (comprising both herbaceous and woody foliage mass) is well documented from analyses covering global drylands[11]. Likewise, global syntheses of the impact on woody vegetation (based on benchmark tree cover maps) evidence the close relationship between rainfall climatology and the density of woody plants[12]. At local scale, it has been found that changing rainfall climatology (more specifically the increase in heavy rainfall events) in drylands has led to increased growth of woody plants[2]. This is because woody plant vegetation does not require an even temporal distribution of rainfall over the wet season to the same degree as herbaceous vegetation and is able to use water from the deeper soil layers[2]. Consequently, altered rainfall climatology supposedly can give woody plants an advantage over herbaceous vegetation with shallow rooted systems[13]. This advantage might also be associated with a decrease in the susceptibility of juvenile woody vegetation to the controls induced by fires and herbivores[3].

A global increase in rainfall extremes has been observed in recent decades[4,14], which may be expected to affect the structure and functioning of terrestrial ecosystems[2,15,16], including water balance[15], vegetation productivity[17] and ecosystem composition[16]. Yet, our current knowledge on the impact of rainfall extremes on the competition between woody and herbaceous plants is based on only a few plot-based experimental studies[2,18,19], as it is challenging to assess woody vegetation trends from global long-term optical satellite imagery systems. Recently, a few studies exploited the complementary use of optical and passive microwave satellite data to study temporal changes in woody cover[20,21]. However, no global scale study has yet documented temporal long-term changes in both woody vegetation and green vegetation production and examined the underlying controls, including an analysis of changing rainfall climatology.

Here, we use satellite data of rainfall, $LAI_{max}$ and woody cover for global tropical savannas (see Methods) and find a higher increase in woody vegetation as compared with the increase in $LAI_{max}$. We further observe a widespread declining sensitivity of $LAI_{max}$ to annual rainfall for tropical savannas. These results suggest that altered hydrological conditions have caused a shift in the coexistence of woody and herbaceous vegetation in savanna ecosystems.

## Results

### Changes in $LAI_{max}$ and woody cover

Here, we used the growing season maximum leaf area index ($LAI_{max}$) derived from optical satellite data to study changes in savanna photosynthetic primary production. $LAI_{max}$ is an indicator of the total green production[22], including the foliage mass from both herbaceous and woody vegetation (see Supplementary Fig. 1 for tropical savanna area coverage). We documented a global increase in $LAI_{max}$ over tropical savannas from 1992 to 2012 (Fig. 1a). Annual minimum vegetation optical depth ($VOD_{min}$), which has been shown to be closely related to woody cover in both drylands and humid regions[21], was used as a proxy for woody vegetation. $VOD_{min}$ senses the water content of all woody vegetation components, including the stem and branches, and the impact of herbaceous plants on the $VOD_{min}$ signal is weak[21]. We found that $VOD_{min}$ estimated woody cover increase was significantly higher (expressed by z-scores) than the increase in $LAI_{max}$ ($p < 0.05$), especially for areas with a mean annual rainfall less than 1000 mm yr$^{-1}$ (Fig. 1a and Supplementary Figs. 2 and 3) (spatial distribution of global mean annual rainfall is shown in Supplementary Fig. 4). The increase in $LAI_{max}$ was evenly distributed across rainfall regimes above 500 mm yr$^{-1}$, whereas woody cover increases were strongest for arid/semi-arid regions (Fig. 1a, b). An increase in annual rainfall was observed for all humidity zones, with relatively higher changes in the drier environments (Fig. 1b). A decline in woody cover was observed for high-rainfall zones (>1800 mm yr$^{-1}$), presumably caused by deforestation[21], but also higher rainfall in moist tropical drylands can lead to reduced productivity[23]. We further analysed changes in the sensitivity of $LAI_{max}$ to annual rainfall ($\beta_{LAI\text{-}Rainfall}$), expressed by the slope of a linear regression between $LAI_{max}$ and annual rainfall over a 15-year moving-window period. We found that a majority (56%) of the global savanna areas experienced a reduced sensitivity over the period 1982–2015, and 68% of these areas (38% of global savannas) coincided with areas of increased woody cover. More specifically, the higher the change in annual rainfall and increase in woody cover, the stronger the observed decrease in sensitivity between $LAI_{max}$ and annual rainfall (Fig. 1c). In particular, the semi-arid region showed a predominance of areas of positive trends in woody cover and a reduced sensitivity between $LAI_{max}$ and annual rainfall, whereas for arid and humid regions the distribution of areas with reduced and increased sensitivity of $LAI_{max}$ to annual rainfall was more balanced (being most pronounced for humid regions) (Fig. 1d–f).

### Rainfall climatology driving changes in woody cover

Taken together, our results suggest that a shift in ecosystem structure has occurred under changed rainfall conditions in tropical savanna areas during recent decades. The relative weights of the underlying factors controlling the observed increase in woody cover (Fig. 2a) were determined by a bootstrapping technique based on the LMG (Lindeman, Merenda and Gold) method[24]. We found that factors related to inter-annual dynamics in rainfall climatology (including annual rainfall, heavy rainfall frequency and rainy days) were primary factors controlling changes in woody cover with relative weights of 47% for all tropical savannas, 78% in arid, 67% in semi-arid and 36% in humid regions (Fig. 2a). Factors related to changes in air temperature and solar radiation, human management (human population density and fire changes) as well as static biotic- and abiotic-site conditions (mean annual rainfall, rainfall variability, soil organic carbon, elevation and sand fraction) together accounted for the remaining relative weights (53% for all tropical savannas, 22% in arid, 33% in semi-arid and 64% in humid savannas). Change in rainfall intensity was excluded from the analysis due to its close relationship with change in annual rainfall (Supplementary Table 1). For comparison, lower contributions of all explanatory factors to changes in $LAI_{max}$ were observed (Supplementary Table 2). The contribution of each explanatory variable to changes in woody cover is shown in Supplementary Fig. 5. Although grazing has shown to impact on the dynamics of woody cover in Southern Africa[25], the lack

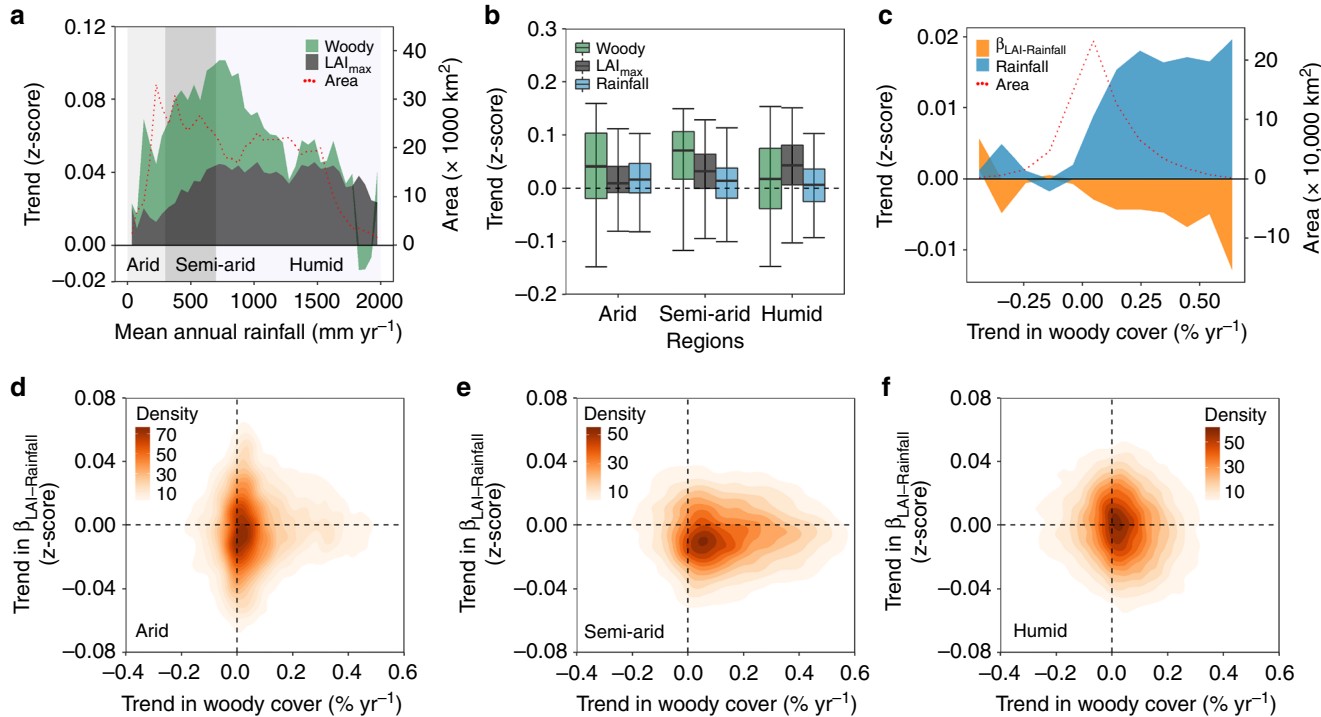

**Fig. 1** Changes in the relationship between vegetation and annual rainfall. **a** Changes in normalized woody cover and $LAI_{max}$ along a mean annual rainfall gradient (50 mm yr$^{-1}$ steps) for 1992–2012. Arid, semi-arid and humid regions were defined as < 300, 300–700 and > 700 mm yr$^{-1}$, respectively. **b** Comparison of changes in normalized woody cover, $LAI_{max}$ and annual rainfall for arid ($n = 3853$), semi-arid ($n = 8159$) and humid regions ($n = 15,824$). **c** Changes in the sensitivity of $LAI_{max}$ to annual rainfall (expressed by the linear regression slope over 15 years: $\beta_{LAI-Rainfall}$) and normalized annual rainfall for 1982–2015 along with changes in woody cover (10% yr$^{-1}$ steps). Normalizations were calculated by the z-score: $X_i = (x_i - x)/\sigma(x)$ (where $x$ and $\sigma(x)$ are the mean and standard deviation of $x_i$). **d–f** Spatial agreement of areas with changes in woody cover and areas with changes in $\beta_{LAI-Rainfall}$ for arid, semi-arid and humid regions. Lines in **b** from top to bottom represent the maximum, third quartile, median, first quartile and minimum values

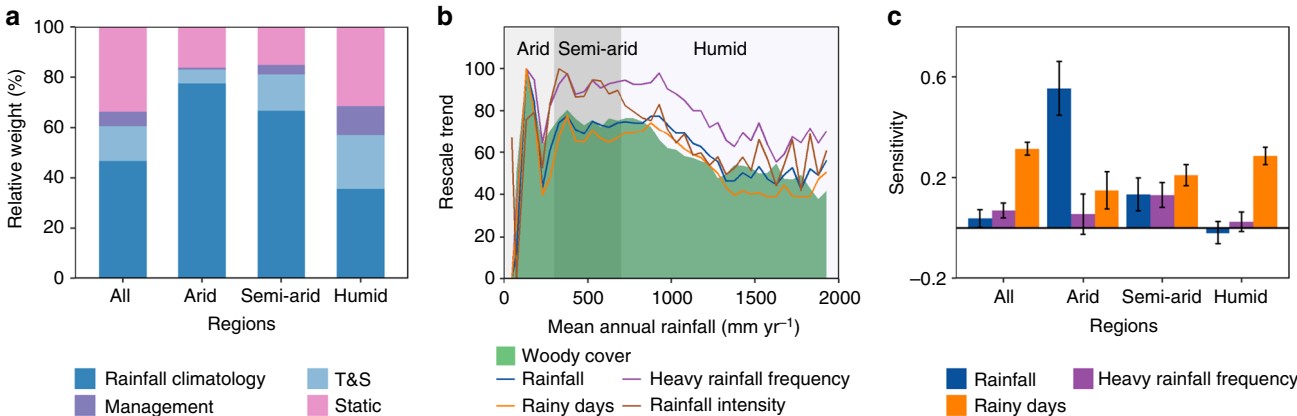

**Fig. 2** Drivers of changes in woody cover in tropical savannas. **a** Relative weights of variables describing rainfall climatology as well as environmental variables explaining woody cover changes (1992–2012) from the LMG method based on a least squares regression model. Rainfall climatology: changes in annual rainfall, heavy rainfall frequency and rainy days; T&S: changes in mean annual air temperature and solar radiation (represented by cloud cover); management: changes in human population density and fire (burned area fraction); static includes variables: mean annual rainfall, coefficient of inter-annual rainfall variability, soil organic carbon, elevation and sand fraction. **b** Changes in rainfall climatology and woody cover were averaged along the mean annual rainfall gradient (50 mm yr$^{-1}$ steps); all variables were rescaled to 0–100. **c** Sensitivity of changes in woody cover towards changes in rainfall variables (expressed as slope from a linear regression; $\beta_{woody-Rainfall}$) for each rainfall region. Error bars indicate the 5–95 % confidence intervals. All rainfall climatology variables were normalized

of reliable global data on grazing intensities impeded this factor to be included in the model.

We further examined the relationship of changes in different rainfall variables (representing rainfall climatology) and changes in woody cover. We found a similar pattern between changes in woody cover and changes in rainfall variables along the entire rainfall gradient, with the strongest changes happening in arid and semi-arid regions (Fig. 2b). The sensitivity of changes in

woody cover to changes in rainfall climatology was assessed by the slope of a linear regression between changes in woody cover and changes in rainfall variables ($\beta_{woody-Rainfall}$) (annual rainfall, heavy rainfall frequency and rainy days) with results varying across rainfall regimes (Fig. 2c). For arid savannas, woody cover changes were most sensitive to changes in annual rainfall and rainy days. In semi-arid savannas, the sensitivity of changes in woody cover to annual rainfall, heavy rainfall frequency and rainy days was of almost equal size, while in the humid region, woody cover was most sensitive to number of rainy days. The sensitivity of each explanatory variable to changes in woody cover is shown in Supplementary Table 3. These findings were further supported by a partial regression analysis that produced comparable results (Supplementary Table 4).

**Spatial distribution of ecosystem structure changes**. The reduced sensitivity of $LAI_{max}$ to annual rainfall was evident over all continents (Fig. 3a) with large clusters of reduced sensitivity in the Sudano–Sahelian zone, in southern Africa and northern Australia. Scattered areas showing an increased sensitivity of $LAI_{max}$ to annual rainfall were also found across all continents (for example woodlands in southern Africa, humid zones of western Africa, South America and western Australia). As increased heavy rainfall frequency has been documented to favour woody vegetation[2], supported by a biophysical explanation of the phenomenon[5], we analysed the impact of heavy rainfall frequency on changes in woody cover (changes in all variables describing rainfall climatology are shown in the supplementary Fig. 6). The spatial correspondence of changes in woody cover and heavy rainfall frequency is shown in Fig. 3b, illustrating that 24% of global savanna regions experienced an increase in both woody cover and heavy rainfall frequency (results for annual rainfall, rainy days and rainfall intensity are shown in the Supplementary Fig. 7). Hot spot areas of change matched the patterns of Fig. 3a and coincided with areas where woody encroachment

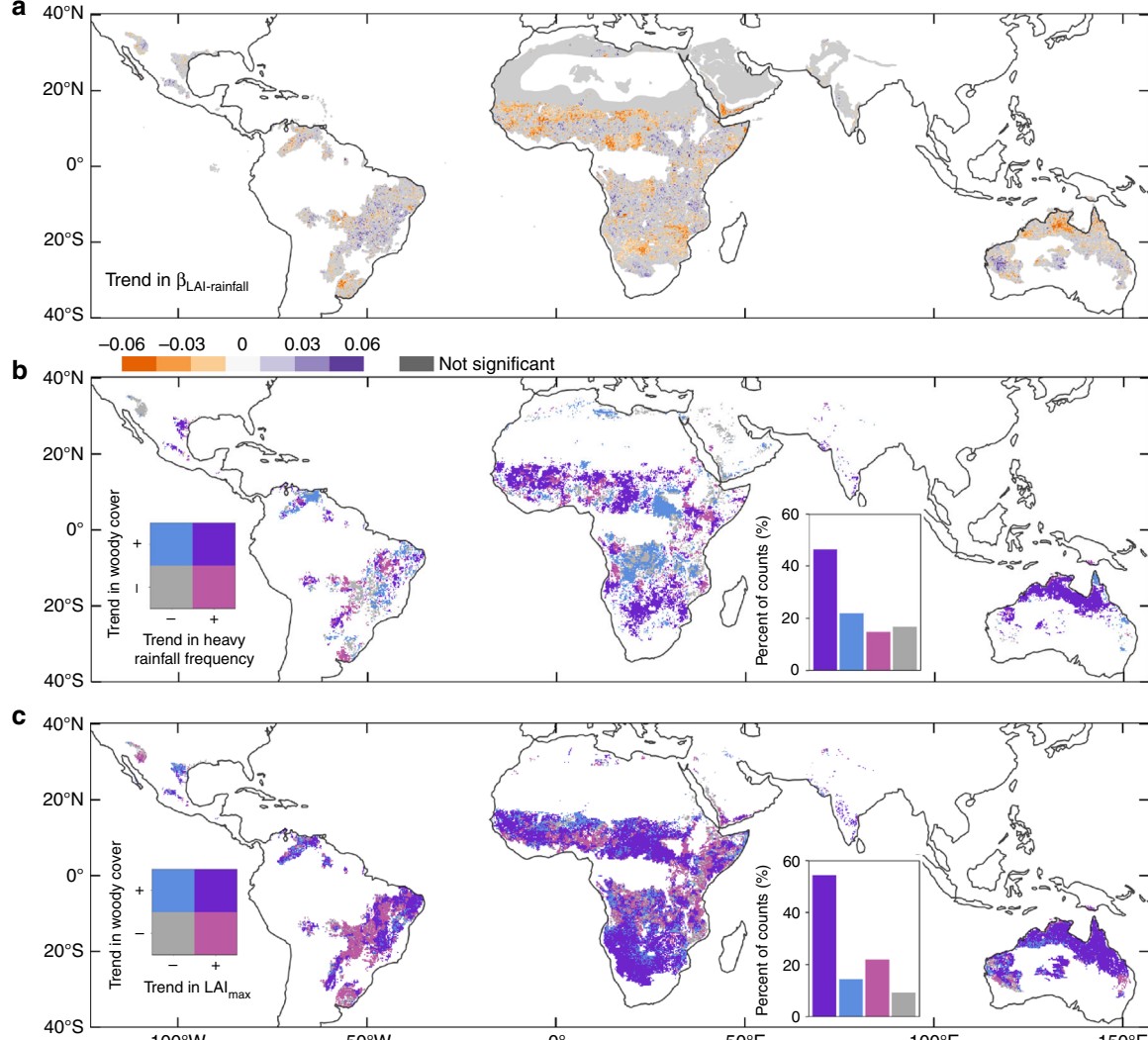

**Fig. 3** Spatial distribution of ecosystem structure changes in tropical savannas. **a** Significant ($p < 0.05$) linear trends in $\beta_{LAI-Rainfall}$ (1982–2015). **b** Spatial agreement of trends in woody cover (1992–2012) and heavy rainfall frequency (1982–2015). Purple areas show increased trends in both heavy rainfall frequency and woody cover, light red areas show an increase in heavy rainfall frequency, but a decrease in woody cover, blue areas show an increase in woody cover but a decrease in heavy rainfall frequency, and grey areas show a decrease in both woody cover and heavy rainfall frequency. The bar plot shows the percent of count for each group with the same colour scheme as the map (total number of pixels = 17,717). **c** Spatial agreement of trends in woody cover (1992–2012) and $LAI_{max}$ (1982–2015). Purple areas show increased trends in both woody cover and $LAI_{max}$, light red areas show a decrease in woody cover and increase in $LAI_{max}$, blue areas an increase in woody cover and decrease in $LAI_{max}$ and grey areas show a decrease in both variables

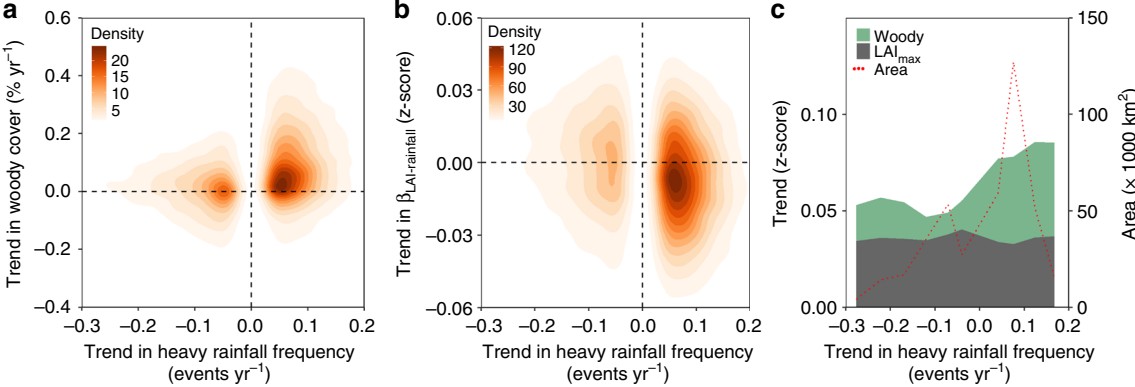

**Fig. 4** Relationship between vegetation and heavy rainfall frequency in tropical savannas. **a** Spatial agreement of areas of changed woody cover and areas of changed heavy rainfall frequency. **b** The same for areas of changes in $\beta_{LAI\text{-}Rainfall}$ and areas of changed heavy rainfall frequency. **c** Changes in normalized woody cover and $LAI_{max}$ for 1992–2012 along a gradient of changes in heavy rainfall frequency (0.05 events yr$^{-1}$ steps)

has been documented by field studies and satellite data[5,7,18,19]. Areas with an increasing woody cover but decreasing heavy rainfall frequency were located in African woodlands north and south of the equatorial rainforest and may be related to non-symmetric responses to wet years or changes in human management[26].

Most areas with an increased woody cover also experienced an increase in $LAI_{max}$ (54%) (Fig. 3c). However, areas of increased $LAI_{max}$ and a decreased woody cover were found in the Brazilian cerrado and Nigeria, which spatially agree with areas of large-scale forest to farmland conversions[27,28]. For areas of both increased woody cover and $LAI_{max}$, particularly in Southern Africa, Northern Australia and Central African Republic large clusters of pixels showing higher trends in woody growth as compared with $LAI_{max}$ is observed (Supplementary Fig. 3) consistent with the substantial woody encroachment reported in refs. [5,6].

The majority of areas where heavy rainfall frequency increased also showed an increased woody cover (Fig. 4a), and areas of increased heavy rainfall frequency also aligned with areas of a decreased sensitivity of $LAI_{max}$ towards annual rainfall (Fig. 4b). The magnitude of increases in heavy rainfall frequency further appeared to be associated with the magnitude of increased woody cover, which was not the case for changes in $LAI_{max}$ (Fig. 4c).

## Discussion

Our study shows global increases in $LAI_{max}$ and annual rainfall, in particular in arid and semi-arid regions. This increase implies that the total production of green vegetation is increasing in recent decades, being in line with studies by Zhu[29] and Fensholt[11]. However, at the same time, the sensitivity of $LAI_{max}$ to annual rainfall declined over the period analysed, supporting that a shift in the relationship between vegetation productivity and rainfall is taking place[30,31]. As $LAI_{max}$ comprises both herbaceous and woody foliage production, the decreased sensitivity between $LAI_{max}$ and annual rainfall suggests a change in the ecosystems structure, and/or changes in the rainfall climatology. It has been shown that increases in rainfall benefit herbaceous plants at the expense of woody plant growth[18], whereas woody plant pro-liferation often occurs in periods following lower rainfall, when the competition with herbaceous plants is low[32]. However, the decreased sensitivity between $LAI_{max}$ and annual rainfall does not point towards a strongly increased herbaceous mass, since her-baceous vegetation (and also woody foliage production) in savannas reacts mainly to the rainfall conditions of a given year[33]. At the same time, we find that increases in woody vegetation (as measured by the water content of the entire woody plant) were

higher as compared with increases in $LAI_{max}$, and this was, at global scale, largely controlled by rainfall climatology. Increases in woody cover were mainly found in areas with a decreasing sensitivity between $LAI_{max}$ and annual rainfall, suggesting that woody vegetation benefits more from the increased annual rainfall as compared with $LAI_{max}$. A possible explanation is found in the altered rainfall climatology (distribution of annual rainfall). Here, we found a strong relationship between increases in woody cover and increases in the frequency of heavy rainfall events, which has also been demonstrated at experimental sites[2,19]. Since herbaceous vegetation is known to be vulnerable to an uneven seasonal distribution of rainfall[34], altered rainfall climatology is likely to affect the coexistence of woody and herbaceous vegeta-tion, e.g., favouring the expansion of the ranges of woody plants[2,13]. This effect would imply that large-scale woody encroachment reported in savanna ecosystems[2] is at least partly driven by changes in the rainfall regime. This further implies a pronounced change in the functioning of global savanna eco-systems within the last 30 years, which would have a profound impact on the biodiversity, carbon-storage capacity and provi-sioning of these ecosystems.

Moreover, the results may contribute explaining the phenom-enon of the greening of global drylands[11,29]. A strong influence of El Niño/La Nina events on vegetation has been reported in several regions of the world, notably via the impact on rainfall[13,16]. From a wider climatic perspective, the increasing frequency of La Niña/El Niño events with global warming suggested by some recent studies[35,36] would lead to an increase in the frequency of heavy rainfall episodes and droughts[37], which could further favour woody vegetation. Experimental studies have demonstrated that increased $CO_2$ positively affected vegetation productivity, parti-cularly the growth of woody plants[9,38]. However, the contribution of $CO_2$ fertilization on the global greening and large-scale woody encroachment is difficult to quantify and global evidence based on observational data has not yet been established[5,9,10,39,40]. Our results do not exclude the possibility of a "$CO_2$ effect", but sug-gest, from a global perspective, woody cover increases in tropical savannas to be influenced by altered rainfall climatology, including a higher annual rainfall, higher frequency of extreme rainfall events and number of rainy days induced by climate change.

## Methods

**Climatic data.** We used monthly climatic data (rainfall, temperature and cloud cover) with a spatial resolution of 0.5° (1982–2015) from the University of East Anglia's Climate Research Unit CRU TS 4[41] and CHIRPS satellite-based daily rainfall estimates at 0.05° spatial resolution[42], both resampled to 0.25° (bilinear and nearest neighbour, respectively). CRU rainfall was used to calculate the relationship

between LAI and annual rainfall. Heavy rainfall frequency were extracted from CHIRPS using the number of rainy days ($\geq 1$ mm day$^{-1}$) for 1982—2015 above the 90th percentile of observed daily rainfall, the number of rainy days and rainfall intensity were calculated for each year.

**Vegetation data**. We used the latest version of GIMMS3g LAI v1 that was generated by training the GIMMS3g v1 NDVI (Normalized Difference Vegetation Index) with MODIS LAI using an artificial neutral network[43]. This dataset was provided with a bi-monthly temporal resolution in a 1/12° spatial resolution from 1982 to 2015. LAI$_{max}$ is an indicator of the total green vegetation production[22], including the foliage mass from both herbaceous and woody vegetation. The growing season maximum leaf area index (LAI$_{max}$) was used to study changes in savanna photosynthetic primary production. The LAI$_{max}$ was resampled to a 0.25° spatial resolution using nearest neighbour interpolation to match the VOD pixel size.

Contrary to LAI, vegetation optical depth (VOD) is sensitive to the water content of both the green and non-green woody parts of vegetation[44] and has been shown to better reflect the woody vegetation signal[20]. The data were retrieved from satellite passive-microwave observations and several merged sensors at a spatial resolution of 0.25° from 1992 to 2012[45]. We followed the approach of Brandt et al.[21] who found a strong linear correlation between annual minimum VOD values and woody cover over Africa ($r^2 = 0.81$, slope $= 64.46$, regression forced through 0 to avoid negative values) and transformed the annual minimum VOD values to the unit woody cover (%) by multiplication with 64.46. By using the seasonal minimum, we also minimized the influence of annual herbaceous vegetation and avoided saturation effects over dense forests[21].

**Environmental data**. Additional datasets, including dynamic environmental/ demography variables (changes in air temperature, solar radiation, human population density and burned area fraction) and static environmental variables (mean annual rainfall, coefficient of inter-annual rainfall variability, soil organic carbon, elevation and sand fraction), were used to study the relationship between changed herbaceous/woody vegetation and rainfall climatology changes. Trends in air temperature and solar radiation (represented by cloud cover) for 1992–2012 were calculated using the CRU TS4 datasets[41]. The differences in human population density between 2015 and 2000 were based on the Gridded Population of the World (GPWv4)[46] dataset. Mean annual burned fraction for 1997–2012 from the Global Fire Emissions Database (GFED4s)[47] was used to analyse the impact of changes in fire activity. Soil organic carbon (SOC) and sand fraction were extracted from the Harmonized World Soil Database (HWSD v1.2)[48]. The elevation extracted from the Shuttle Radar Topography Mission data (SRTM 90 m)[49]. All datasets collected were resampled to a 0.25° spatial resolution using nearest neighbour interpolation (except GPWv4 and GFED4s that were provided in 0.25° spatial resolution).

**Additional data**. We defined the areal extent of tropical savannas based on the ecoregion classification by Olson et al.[50]. To minimize the human footprint on the analyses conducted, global land cover (GLC-SHARE) data[51] were used to mask out cropland areas with a cropped percentage above 70%. ESA CCI land cover (https:// www.esa-landcover-cci.org/) from 2010 was used to mask irrigated areas. Additionally, ESA CCI land cover was used to limit our analysis to savanna vegetation by masking out water areas.

**Analysis**. We performed a linear regression between herbaceous vegetation (LAI$_{max}$) and annual rainfall over a 15-year moving window to derive the beta coefficient ($\beta_{LAI-Rainfall}$), which is frequently used as a measure of the sensitivity of vegetation to variation in rainfall. Variables were normalized at each grid point by the inter-annual standard deviation in each study period prior to linear regression analysis. A non-parametric Theil–Sen trend analysis was then applied to detect trends in $\beta_{LAI-Rainfall}$ and the Mann–Kendall (MK) test was applied to evaluate $\beta_{LAI-Rainfall}$ trends at the 95% significance level ($p < 0.05$) taking temporal autocorrelation into account. Trends in woody cover and rainfall climatology variables were also calculated using this method. Here, the MK significance test based on linear trends was avoided as a linear response of changes in woody cover to changes in rainfall climatology variables was not to be expected. Moreover, due to the high inter-annual variability of climatology, significant trends could not be expected[52]. Welch's unequal variances $t$ test was used to test for significant difference between temporal changes in LAI$_{max}$ and woody cover.

The relative weights of the controlling factors on changes in woody cover were estimated using the LMG method (a bootstrapping technique)[24], which determines the explaining power of each variables in the model as a share of 100%. A least squares regression model and a partial regression model were used to relate changes in woody cover to changed rainfall climatology variables and environmental variables. Changed woody cover was used as the response variable. The explanatory variables included: (1) change in annual rainfall; (2) change in heavy rainfall frequency; (3) change in rainy days; (4) change in annual air temperature; (5) change in solar radiation; (6) change in human population density; (7) change in burned area fraction; (8) mean annual rainfall; (9) coefficient of inter-annual rainfall variability; (10) soil organic carbon; (11) elevation; (12) sand fraction. The model was run for tropical savannas (All) and its sub-regions

including arid ($< 300$ mm yr$^{-1}$), semi-arid (300–700 mm yr$^{-1}$) and humid regions ($> 700$ mm yr$^{-1}$) divided according to mean annual rainfall during 1992–2012. Each variable $x_i$ was normalized as $X_i = (x_i -x)/\sigma(x)$ (where $x$ and $\sigma(x)$ are the mean and standard deviation of $x_i$) before modelling the data and all analysis were conducted in R.

**Reporting summary**. Further information on experimental design is available in the Nature Research Reporting Summary linked to this article.

## Data availability
GIMMS3g LAI data are available at http://cliveg.bu.edu/modismisr/lai3g-fpar3g.html. The VOD data is provided by Y. Liu, Nanjing University of Information Science & Technology, China. CRU TS 4 datasets (rainfall, temperature and cloud cover) are available from the University of East Anglia (http://www.cru.uea.ac.uk/). CHIRPS datasets can be downloaded from ftp://ftp.chg.ucsb.edu/pub/org/chg/ products /CHIRPS-2.0. The GFED4s datasets are available at http://www.globalfiredata.org/data.html. The Gridded Population of the World data can be downloaded from http://sedac.ciesin. columbia.edu/data/collection/gpw-v4. The soil maps are available at http://worldgrids. org/doku.php/wiki:layers. The SRTM 90 is available at http://www.cgiar-csi.org/data/ srtm-90m-digital-elevation-database-v4-1.

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

## Acknowledgements

This study is jointly supported by the China Scholarship Council (CSC, 201506190076) and the Danish Council for Independent Research (DFF) project: Greening of drylands (DFF-6111-00258): Towards understanding ecosystem functioning changes, drivers and impacts on livelihoods. J.P. acknowledges the financial support from the European Research Council Synergy grant ERC-SyG-2013-610028 IMBALANCE-P. F.G. acknowledges funding from the NERC/DFID Future Climate For Africa programme under the AMMA-2050 project (Grant NE/ M019950/1). M.B. was supported by the AXA postdoctoral research grant. Also, we would like to thank Y.Y. Liu for providing the VOD data.

## Author contributions

W.Z., M.B., R.F. and J.P. designed the study. W.Z. conducted the analyses with support by M.B., R.F., J.P., F.G., X.T. and F.T. W.Z., M.B. and R.F. drafted the paper with contributions by J.P. All authors discussed the results and commented on the paper.

## Additional information

**Competing interests:** The authors declare no competing interests.

