## [Peer Review File · Nature Communications]

Reviewers' comments:

Reviewer #1 (Remarks to the Author):

This paper presents a set of analyses that provide evidence for global shifts in the structure of dryland vegetation. These shifts are inferred from changing patterns of LAImax (for grasses) and VOD minima (for trees), and the overwhelming pattern is one of increased woody cover driven by increasing amounts of rainfall.

While previous work by the authors and other has integrated VOD and optical vegetation indices to tease out structure of vegetation (Guan et al., 2012), the novelty of this work comes from both the spatial extent and timescales of the analyses. This is truly one of the first efforts to tackle global shift in tree/grass structure and certainly the first to so clearly tie these changes to altered climatology.

I am quite positive about this work, and - having participated in a review of a previous version of this manuscript in a different venue - I feel that it has been strengthened in terms of its presentation of both the figures and the text. Figure 2a is the key result that justifies the strongly worded title and conclusions. These results show clearly that changing patterns of rainfall are most strongly related to the shifts in vegetation structure observed across tropical drylands, and that result is important and one hopes will greatly affect the community of researchers working on vegetation change in these regions.

Minor points:

1. It's hard to tease out of the presentation, but the authors find a decline in woody vegetation at high rainfalls (fig. 1a) despite the fact that these areas are also seeing increases in rainfall (fig 1c; appears that locations with negative woody cover trends also have positive rainfall trends). They put this result onto deforestation, which it certainly may be. But other work (Guan et al., 2013) has proposed that higher rainfall in moist tropical drylands could lead to reduced productivity. Certainly there are land use transformations happening across drylands at all rainfall gradients. It's possible that they are most pronounced at wetter levels of rainfall, but also possible that the trends observed arise from other interactions between plant structure, function, and climate.

2. I really like Figure 3b, both in terms of how it is presented and what it is saying about the role of rainfall extremes in vegetation structure. However, there is not a similar figure for total rainfall. The authors are clear in places about the fact that these two trends (total rainfall and rainfall extremes) are confounded (e.g. L115-117), but they put the total rainfall results in the supplemental and highlight these results instead. I don't see the rationale for this decision; although I admit that the "extreme rainfall" story is stickier and resonates with other work, the "increased rainfall" story seems more parsimonious.

3. Figure 3a shows changing sensitivity of LAImax to rainfall, which is inferred to mean reductions in grass cover. While the text states that the map of declining grass cover generally corresponds to increases in tree cover, a visual comparison of Figures 3a and 3b also reveals areas where this inverse relationship does not hold. A quadrant map of grass increases/decrease and woody cover increase/decrease could be interesting?

In summary, I find this work of high value to the community of scholars working on climate/vegetation relationships in drylands and of sufficient scope and ambition to ensure it's widespread interest. The work is clearly presented and the methods are well-presented.

As with my prior review, I encourage the authors to contact me if they feel the need for clarification on any of these points.

Kelly K. Caylor
caylor@ucsb.edu

References:

Guan et al., 2012: <https://doi.org/10.1016/j.rse.2012.06.005>

Guan et al. 2013: <https://doi.org/10.1890/ES12-00232.1>

Reviewer #2 (Remarks to the Author):

General

I have enjoyed reading this paper as it offers interesting and novel insights into the global drivers of woody cover change. As far as I know it is one of the few papers that attempts to quantify the causes of woody cover change across the savannas of the world. I find the analysis neat and innovative and I particularly enjoyed reading about the decreasing sensitivity of LAI to rainfall, presumably brought about by encroachment.

On the other hand; my concerns with the paper are 1) It is uncertain to me how a reasonable measure LAI_{max} represents herbaceous cover. Clarity on this topic would certainly benefit the paper. 2) The authors ecological explanation of the causes of these patterns seem to skip out on large parts of the savanna literature and they fail to consider or interpret their findings in light of contradictory papers. This leaves the paper as a showpiece of technical expertise and impressive analysis instead of a paper that advances ecological understanding to the extent that it could. 3) An additional large concern is that whilst the authors are attempting to understand some of the causes of encroachment and ecosystem structural change they completely fail to acknowledge the potential role of elevated CO₂ in this process. Such a paper can make a valuable contribution and I feel the authors don't do the argument/debate any justice by completely ignoring a widely proposed cause of encroachment. 4) The authors present an analysis on LAI max trends over time but then don't really ever refer to it again. It is an intriguing result and it is not clear why its relevance is ignored.

Detailed comments

Line 37: If you going to discuss the causes then its fair to present all the potential suggested causes which other than herbivores and fire includes increasing CO₂.

Line 49: ref?

Line 64: This explanation is somewhat problematic for me. How strongly influenced? This assumption/statement forms the basis of the analysis. I have gone back to the original paper am cannot quite see how strong and reliable this relationship is. Certainly I understand that in the drier regions you can essentially "ignore" the tree cover component's contribution to LAI as tree cover density is so much lower. But then in this analysis you included the arid shrublands of Somalia and Ethiopia which are arid yet essentially grassless. I really do need some clarity on this issue. Furthermore, to allow the readers to assess the value of this measure it would be very helpful to include not only a map of LAI_{max} change but also a map of LAI_{max} so the readers can assess for themselves how reasonable an approximation this value is?

Line 74: Could this increase in LAI not be driven by an increase in shrubs?

Line 119: What about changes in fire?

Line 152: There is barely any grass in this much of this region, especially in the horn of Africa. So its not too surprising there is not a lot of sensitivity here. Would this analysis not benefit by excluding thickets in the analysis (see comments in supp material fig 1)

Line 164: I find it remarkable that the authors so steadfastly ignore the elevated CO2 hypothesis in their interpretation of their results

Line 195: But how do you tie in this explanation with the very well established pattern of increasing grass biomass with increased rainfall? All your rainfall metrics show some of this pattern. I think demonstrating the decline of sensitive to LAI_{max} to rainfall is elegant and points really nicely to the impacts of closed canopies. However the authors paint the trend in woody cover change as very much a rainfall story, and I find their ecological argument to not be very convincing. Firstly increasing rainfall undoubtedly increases grass biomass. Increases in rainfall have also been shown to disproportionately benefit grass at the expense of tree growth (see February et al). There is also papers that suggest woody plant proliferation occurs at the end of droughts, where the lack of grass favours tree establishment (e.g. Scholes and Archer 1997; Bond 2007, Riginos 2009). This is a large gap in your argument.

February, E. C., Higgins, S. I., Bond, W. J., & Swemmer, L. (2013). Influence of competition and rainfall manipulation on the growth responses of savanna trees and grasses. *Ecology*, 94(5), 1155-1164.

Line 210: Obviously limited by time and energy - but this analysis would be incredibly insightful if it was complemented by a similar analysis using fire.

Following on from your title which says you talking about changes in ecosystem structure, you observe an interesting trend in the increase of grass biomass, yet you don't mention it further. This is also an intriguing result. You seem to identify two processes 1) uncoupling of rainfall and grass biomass relationship presumably caused by the shading out of grasses by trees and 2) areas where an increase in grass biomass is occurring. Some discussion as to what is causing this trend would be very interesting.

In light of recent research by Reich et al (P. B. Reich, S. E. Hobbie, T. D. Lee, M. A. Pastore, Unexpected reversal of C3 versus C4 grass response to elevated CO2 during a 20-year field experiment. *Science* 360, 317–320 (2018). doi:10.1126/science.aas9313pmid:2967459) they suggest that we could be expecting to see c4 grass benefiting from elevated co2. Your results do indicate this trend - it would be of interest to hear how your interpretation of results fits into this debate?

Reply to reviewers' comments:

Dear editor and reviewers. We thank you all very much for being in support of our work and for providing such pertinent and constructive comments. We agree with the points raised and have made revisions to the manuscript accordingly. The major changes in the revised version are:

- (1) Now we have made clear that the vegetation metric LAI_{max} comprises woody, herbaceous and shrub vegetation to make our expression more accurate in the revised version. We agree with Reviewer 2 that even though the contribution from woody vegetation on LAI_{max} trends is expected to be quite low in arid and semi-arid areas, we can not exclude a possible influence from the woody/shrub stratum. This clarification of the terminology, however, does not change our results as we still have two different vegetation metrics; VOD being only sensitive to the entire woody plants and LAI_{max} being sensitive to the entire green vegetation stratum (please see further explanation below).
- (2) We have now elaborated on the interpretations of our findings related to changes in LAI_{max} .
- (3) We have included a discussion of the role of CO_2 fertilization on dynamics of woody vegetation.

Reviewers' comments:

Reviewer #1 (Remarks to the Author):

This paper presents a set of analyses that provide evidence for global shifts in the structure of dryland vegetation. These shifts are inferred from changing patterns of LAI_{max} (for grasses) and VOD minima (for trees), and the overwhelming pattern is one of increased woody cover driven by increasing amounts of rainfall.

While previous work by the authors and other has integrated VOD and optical vegetation indices to tease out structure of vegetation (Guan et al., 2012), the novelty of this work comes from both the spatial extent and timescales of the analyses. This is truly one of the first efforts to tackle global shift in tree/grass structure and certainly the first to so clearly tie these changes to altered climatology.

I am quite positive about this work, and - having participated in a review of a previous version of this manuscript in a different venue - I feel that it has been strengthened in terms of its presentation of both the figures and the text. Figure 2a is the key result that justifies the strongly worded title and conclusions. These results show clearly that changing patterns of rainfall are most strongly related to the shifts in vegetation structure observed across tropical drylands, and that result is important and one hopes will greatly affect the community of researchers working on vegetation change in these regions.

We thank the reviewer very much for supporting our work. We agree with the points raised and have revised the manuscript accordingly. We have worked a lot on this manuscript since the first submission, and we think it has been improved substantially, especially thanks to your help.

Point-by-point answers:

Minor points:

1. It's hard to tease out of the presentation, but the authors find a decline in woody vegetation at high rainfalls (fig. 1a) despite the fact that these areas are also seeing increases in rainfall (fig 1c; appears that locations with negative woody cover trends also have positive rainfall trends). They put this result onto deforestation, which it certainly may be. But other work (Guan et al., 2013) has proposed that higher rainfall in moist tropical drylands could lead to reduced productivity. Certainly there are land use transformations happening across drylands at all rainfall gradients. It's possible that they are most pronounced at wetter levels of rainfall, but also possible that the trends observed arise from other interactions between plant structure, function, and climate.

Response: We fully agree that the interactions between plant structure, function and climate are important. We have followed your suggestion and have included the suggested work of Guan et. al., 2013. The new sentence reads: "A decline in woody cover was observed for high rainfall zones ($>1800 \text{ mm yr}^{-1}$), presumably caused by deforestation, but also higher rainfall in moist tropical drylands can lead to reduced productivity".

Reference:

Guan, K. et al. Seasonal coupling of canopy structure and function in African tropical forests and its environmental controls. *Ecosphere* 4, 1–21 (2013).

2. I really like Figure 3b, both in terms of how it is presented and what it is saying about the role of rainfall extremes in vegetation structure. However, there is not a similar figure for total rainfall. The authors are clear in places about the fact that these two trends (total rainfall and rainfall extremes) are confounded (e.g. L115-117), but they put the total rainfall results in the supplemental and highlight these results instead. I don't see the rationale for this decision; although I admit that the "extreme rainfall" story is stickier and resonates with other work, the "increased rainfall" story seems more parsimonious.

Response: As already mentioned by the reviewer, one of the main reasons to decide for heavy rainfall was the agreement with other experimental work, which demonstrated this behaviour at field plot level and was our major motivation to do this study. Here we want to expand on these studies and show that these patterns apply at global scale. Also, annual rainfall alone tells little about the distribution of rainfall, which we think is the key explaining the decoupling between LAI and annual rainfall. Heavy rainfall is part of the annual rainfall, but annual rainfall is not part of the heavy rainfall. Heavy rainfall is thus a variable on its own, and we believe it will be most attractive to illustrate this in the main text, whereas all the other information can be found in the supplementary part.

3. Figure 3a shows changing sensitivity of LAI_{max} to rainfall, which is inferred to mean reductions in grass cover. While the text states that the map of declining grass cover generally corresponds to increases in tree cover, a visual comparison of Figures 3a and 3b also reveals areas where this inverse relationship does not hold. A quadrant map of grass increases/decrease and woody cover increase/decrease could be interesting?

Response: Thank you for these excellent suggestions, which we have included in the revised MS. Note that in this revised version we are less strict with the term "grass cover", since we cannot guarantee that changes in LAI_{max} is exclusively caused by changes in the herbaceous components. We have softened the formulations in this respect and call LAI_{max} rather an indicator for the green vegetation mass, which however does not change anything of the story as we still have two different vegetation metrics; VOD being sensitive to the woody vegetation only and LAI_{max} being sensitive to the entire vegetation stratum (in most savanna areas of course being predominantly influenced by changes in the herbaceous vegetation).

Your suggestion is very interesting and we have added the map in Figure 3. The patterns make sense: decreases in woody cover but increases in LAI_{max} are mainly found in hot spots of human made land use conversion areas, with the largest hot spots in the Brazilian cerrado and Nigeria. We have also added new text describing this phenomenon, line 172-178:” Most areas with an increased woody cover also experienced an increase in LAI_{max} (54%) (Fig. 3c). However, examples of an increased LAI_{max} and a decreased woody cover were found in the Brazilian cerrado and Nigeria, and spatially agree with areas of large scale forest to farmland conversions^{27,28}. For areas of both increased woody cover and LAI_{max}, particularly in Southern Africa, Northern Australia and Central African Republic large clusters pixels showing higher trends in woody growth as compared to LAI_{max} is observed (Supplementary Fig. 3) consistent with the woody encroachment reported in5,6.”. Thanks, this again adds value to our study.

References:

5, Stevens, N., Lehmann, C. E. R., Murphy, B. P. & Durigan, G. Savanna woody encroachment is widespread across three continents. *Global Change Biology* 23, 235–244 (2017).

6, Venter, Z., Cramer, M. D. & Hawkins, H. Drivers of woody plant encroachment over Africa. *Nature Communications* 1–7 (2018).

27, Audu, E. B. Fuel wood consumption and desertification in Nigeria. *International Journal of Science and Technology* 3, 1–5 (2013).

28, Lapola, D. M. et al. Pervasive transition of the Brazilian land-use system. *Nature Climate Change* 4, 27–35 (2014).

In summary, I find this work of high value to the community of scholars working on climate/vegetation relationships in drylands and of sufficient scope and ambition to ensure it's widespread interest. The work is clearly presented and the methods are well-presented.

Response: Thank you again for your support. These excellent remarks have further improved our study.

As with my prior review, I encourage the authors to contact me if they feel the need for clarification on any of these points.

References:

Guan et al., 2012: <https://doi.org/10.1016/j.rse.2012.06.005>

Guan et al. 2013: <https://doi.org/10.1890/ES12-00232.1>

Reviewer #2 (Remarks to the Author):

General

Authors: We thank the reviewer very much for providing pertinent and constructive comments.

1. I have enjoyed reading this paper as it offers interesting and novel insights into the global drivers of woody cover change. As far as I know it is one of the few papers that attempts to quantify the causes of woody cover change across the savannas of the world. I find the analysis neat and innovative and I particularly enjoyed reading about the decreasing sensitivity of LAI to rainfall, presumably bought about by encroachment.

Response: We thank you for the constructive review and the support of our work.

On the other hand; my concerns with the paper are 1) It is uncertain to me how a reasonable measure LAI_{max} represents herbaceous cover. Clarity on this topic would certainly benefit the paper.

Response: We notice that this is a key point and have put effort in the clarification. In fact, although LAI_{max} is dominated by herbaceous cover in most savannah areas, we realised that we have to tone down this relationship between LAI_{max} and the herbaceous layer. Indeed, LAI_{max} will always include a component coming from the woody foliage mass, and we thus now refrain from giving the reader the impression that LAI_{max} is to be interpreted as only being sensitive to changes in the herbaceous vegetation. We introduce LAI_{max} as an indicator for the total green vegetation production, which is controlled by both herbaceous and woody foliage. As reviewer suggested we have provided maps of both long-term avg. % woody cover and LAI_{max} to support the spatial interpretation of the results presented. This clarification does not change any results or the conclusions that we draw, but should hopefully reduce the ambiguity in the interpretation of what we show. Please see details below.

2) The authors ecological explanation of the causes of these patterns seem to skip out on large parts of the savanna literature and they fail to consider or interpret their findings in light of contradictory papers. This leaves the paper as a showpiece of technical expertise and impressive analysis instead of a paper that advances ecological understanding to the extent that it could.

Response: Thanks for making us aware of this. We have restructured the discussion in the light of the suggested changes and literature, and have now interpreted our results including these studies with the aim to also provide an advancement on our ecological understanding. Please see details below.

3) An additional large concern is that whilst the authors are attempting to understand some of the causes of encroachment and ecosystem structural change they completely fail to acknowledge the potential role of elevated CO₂ in this process. Such a paper can make a valuable contribution and I feel the authors don't do the argument/debate any justice by completely ignoring a widely proposed cause of encroachment.

Response: Again we fully agree on this point. We have added the following paragraph in line 40-42: "Recently, increased atmospheric CO₂ concentrations were suggested to increase the growth rate of trees relative to grasses^{5,8-10}." And in the discussion 241-248: "Experimental studies have demonstrated that increased CO₂ positively affected vegetation productivity, particularly the growth of woody plants^{9,38}. However, the contribution of CO₂ fertilization on the global greening and large-scale woody encroachment is difficult to quantify and global evidence based on observational data has not yet been established^{5,9,10,39}. Our results do not exclude the possibility of a "CO₂ effect", but suggest, from a global perspective, woody cover

increases in tropical savannas to be influenced by altered rainfall climatology, including a higher annual rainfall, higher frequency of extreme rainfall events and number of rainy days induced by climate change.”.

4) The authors present an analysis on LAI max trends over time but then don't really ever refer to it again. It is an intriguing result and it is not clear why its relevance is ignored.

Response: Also here we thank you for raising this point. We have added a few more maps, and especially the new Fig 3c and the associated supplementary figure 3 is interesting, as it shows areas where LAI_{max} is increasing and woody cover is decreasing at the same time. The patterns show some remarkable clusters of decreases in woody cover but increases in LAI_{max} found in hot spots of human induced land use conversions, with the largest hot spots in the Brazilian cerrado and in Nigeria. We have added new text describing this phenomenon, line 172-178:”Most areas with an increased woody cover also experienced an increase in LAI_{max} (54%) (Fig. 3c). However, areas of increased LAI_{max} and a decreased woody cover were found in the Brazilian cerrado and Nigeria which spatially agree with areas of large scale forest to farmland conversions^{27,28}. For areas of both increased woody cover and LAI_{max}, particularly in Southern Africa, Northern Australia and Central African Republic large clusters of pixels showing higher trends in woody growth as compared to LAI_{max} is observed (Supplementary Fig. 3) consistent with the substantial woody encroachment reported in5,6.”.

Detailed comments

2. Line 37: If you going to discuss the causes then its fair to present all the potential suggested causes which other than herbivores and fire includes increasing CO2.

Response: Thanks for pointing this out. We now have included the CO₂ as a potential driver and rephrased the sentence. It now reads: “Recently, increased atmospheric CO₂ concentrations were suggested to increase the growth rate of trees relative to grasses^{5,8-10}” (line 40-43).

References:

Stevens, N., Lehmann, C. E. R., Murphy, B. P. & Durigan, G. Savanna woody encroachment is widespread across three continents. *Global Change Biology* 23, 235–244 (2017).

Niinemets, Ü., Flexas, J. & Peñuelas, J. Evergreens favored by higher responsiveness to increased CO₂. *Trends in Ecology and Evolution* 26, 136–142 (2011).

Buitenwerf, R., Bond, W. J., Stevens, N. & Trollope, W. S. W. Increased tree densities in South African savannas: >50 years of data suggests CO₂ as a driver. *Global Change Biology* 18, 675–684 (2012).

Devine, A. P., McDonald, R. A., Quaipe, T. & Maclean, I. M. D. Determinants of woody encroachment and cover in African savannas. *Oecologia* 183, 939–951 (2017).

3. Line 49: ref?

Response: We have now inserted a reference: Kulmatiski, A. & Beard, K. H. Woody plant encroachment facilitated by increased precipitation intensity. *Nature Climate Change* 3, 833–837 (2013).

4. Line 64: This explanation is somewhat problematic for me. How strongly influenced? This assumption/statement forms the basis of the analysis. I have gone back to the original paper am cannot quite see how strong and reliable this relationship is.

Response: We agree on that. Our expressions were indeed not very clear (see response to this point above). In fact LAI_{max} is both, herbaceous mass and woody foliage mass. We now make this clear in the text stating that LAI_{max} is an indicator for the total green vegetation mass in contrast to woody vegetation (which is all woody components captured by VOD). This difference is important for the interpretation of our results.

Certainly I understand that in the drier regions you can essentially "ignore" the tree cover component's contribution to LAI as tree cover density is so much lower. But then in this analysis you included the arid shrublands of Somalia and Ethiopia which are arid yet essentially grassless.

Response: Thanks, this is indeed a good example on why we needed to work on the terminology. There was in fact no reason to claim that LAI_{max} was exclusively influenced by changes in the herbaceous vegetation to draw our conclusions from the results.

I really do need some clarity on this issue. Furthermore, to allow the readers to assess the value of this measure it would be very helpful to include not only a map of LAI_{max} change but also a map of LAI_{max} so the readers can assess for themselves how reasonable an approximation this value is?

Response: To summarize our actions on this point:

- We have revised the terminology saying that LAI_{max} is an indicator for the total green vegetation mass, including herbaceous and woody foliage mass. Changes were made in lines 67-69.
- The entire discussion has been adjusted to this point, also to better guide the reader through the meaning of the variables and also through the interpretation of the results. Please see the new text in lines 209-228.
- We have added a new figure in the main text (Fig. 3c) and in supplementary material (Supplementary Fig. 3) showing trends in LAI_{max} in relation to trends in woody cover. The results are interpreted in the text, line 172-178.
- We have added new figures in the supplementary part, which are included in Supplementary Fig. 3.

5. Line 74: Could this increase in LAI not be driven by an increase in shrubs?

Response: Thanks; given the way LAI_{max} was originally defined we fully understand this question. LAI_{max} could indeed be impacted by shrubs which is in accordance with the LAI_{max} definition in our revised version specifying that LAI_{max} is sensitive to changes in both shrub, woody and herbaceous vegetation. We think that our revised definitions of LAI_{max} and VOD give the reader now a better opportunity to correctly interpret these patterns of change.

6. Line 119: What about changes in fire?

Response: Changes in fire are part of the "Management" group in Figure 2a. The individual contribution is shown in Supplementary Fig. 5. We have now made it clearer in the figure caption (2a) to make it easier to find the impact of fire.

7. Line 152: There is barely any grass in this much of this region, especially in the horn of Africa. So its not too surprising there is not a lot of sensitivity here. Would this analysis not benefit by excluding thickets in the analysis (see comments in supp material fig 1)

Response: The problem is to find thresholds of which regions to include and which not. Here we have used the map created by the world wildlife fund. Even though we understand the reviewer's suggestion, we have

decided not to mask these shrubland-dominated areas, as this does not conflict with our definition of what is captured by the LAI_{max} metric.

Reference: Olson, D. M. et al. Terrestrial Ecoregions of the World : A New Map of Life on Earth. *BioScience* 51, 933–938 (2001).

8. Line 164: I find it remarkable that the authors so steadfastly ignore the elevated CO₂ hypothesis in their interpretation of their results

Response: Thank you for pointing this out. We have now included the following references in the added text discussing the role of CO₂ as a driver (see above):

References:

Buitenwerf, R., Bond, W. J., Stevens, N. & Trollope, W. S. W. Increased tree densities in South African savannas: >50 years of data suggests CO₂ as a driver. *Global Change Biology* 18, 675–684 (2012).

Stevens, N., Lehmann, C. E. R., Murphy, B. P. & Durigan, G. Savanna woody encroachment is widespread across three continents. *Global Change Biology* 23, 235–244 (2017).

Devine, A. P., McDonald, R. A., Quaipe, T., & Maclean, I. M. D. (2017). Determinants of woody encroachment and cover in African savannas. *Oecologia*, 183(4), 939–951.

Higgins, S. I. & Scheiter, S. Atmospheric CO₂ forces abrupt vegetation shifts locally, but not globally. *Nature* 488, 209–212 (2012).

9. Line 195: But how do you tie in this explanation with the very well established pattern of increasing grass biomass with increased rainfall? All your rainfall metrics show some of this pattern.

Response: We have now revised the discussion, including this part. However, please note that we do not say that herbaceous biomass does not increase with rainfall. However, the declining relationship between LAI_{max} and annual rainfall suggests that the response of vegetation growth to annual rainfall dynamics weakens and the fact that woody cover increases faster than LAI_{max} in these areas suggest changes in rainfall climatology as an important factor. Since herbaceous vegetation mainly reacts to the rainfall of a given year, this reduced sensitivity does not point towards a strong increase in herbaceous mass.

I think demonstrating the decline of sensitive to LAI_{max} to rainfall is elegant and points really nicely to the impacts of closed canopies. However the authors paint the trend in woody cover change as very much a rainfall story, and I find their ecological argument to not be very convincing. Firstly increasing rainfall undoubtedly increases grass biomass. Increases in rainfall have also been shown to disproportionately benefit grass at the expense of tree growth (see February et al). There is also papers that suggest woody plant proliferation occurs at the end of of droughts, where the lack of grass favours tree establishment (e.g. Scholes and Archer 1997; Bond 2007, Riginos 2009). This is a large gap in your argument.

Response: We thank the reviewer for this comment and have amended the discussion to better reflect different viewpoints in relation to current understanding of the ecology of grasses and trees. Please see our revised text, line 209 to 220: “This increase implies that the total production of green vegetation is increasing in recent decades, being in line with studies by Zhu²⁹ and Fensholt¹¹. However, at the same time, the sensitivity of LAI_{max} to annual rainfall declined over the period analysed, supporting that a shift in the relation-ship between vegetation productivity and rainfall is taking place^{30,31}. As LAI_{max} comprises both herbaceous and woody foliage production, the decreased sensitivity between LAI_{max} and annual rainfall suggests a change in the ecosystems structure, and/or changes in the rainfall climatology. It has been shown that increases in rainfall benefit herbaceous plants at the expense of woody plant growth¹⁸, whereas woody plant proliferation often occurs in periods following lower rainfall, when the competition with herbaceous

plants is low³². However, the decreased sensitivity between LAI_{max} and annual rainfall does not point towards a strongly increased herbaceous mass, since herbaceous vegetation (and also woody foliage production) in savannas reacts mainly to the rainfall conditions of a given year³³, which also includes the suggested literature. Our main argument is that if the rainfall would have caused a strong increase in herbaceous mass, the decoupling would have been less obvious. This does not exclude an increase in grass biomass (Figure. 3c), but it points towards woody vegetation being the driver of the observed decoupling.

February, E. C., Higgins, S. I., Bond, W. J., & Swemmer, L. (2013). Influence of competition and rainfall manipulation on the growth responses of savanna trees and grasses. *Ecology*, 94(5), 1155-1164.

10. Line 210: Obviously limited by time and energy - but this analysis would be incredibly insightful if it was complemented by a similar analysis using fire.

Response: Indeed fire is an important driver, but it did not play a dominant role in our analyses (Fig 2, Fig. S5). This could be related to the coarse pixel resolution (25x25 km) of our VOD data. As stated in Reference : "At a local level, changes in precipitation, burning regimes or herbivory could be driving woody encroachment, but are unlikely to be the explanation of this continent wide phenomenon." In our work we challenge their interpretation of the role of changes in rainfall by looking into changes in rainfall climatology.

Reference:

Devine, A.P., McDonald, R.A., Quaife, T. and Maclean, I.M., 2017. Determinants of woody encroachment and cover in African savannas. *Oecologia*, 183(4), pp.939-951

11. Following on from your title which says you talking about changes in ecosystem structure, you observe an interesting trend in the increase of grass biomass, yet you don't mention it further. This is also an intriguing result.

Response: Please notice that our revised definition of LAI_{max} stating that trends in LAI_{max} should not exclusively be attributed to herbaceous cover. But we agree that the trends in this variable are very interesting and we have added new figures and text to the main text and supplementary part to strengthen this (please see point 9 above).

You seem to identify two processes 1) uncoupling of rainfall and grass biomass relationship presumably caused by the shading out of grasses by trees and 2) areas where an increase in grass biomass is occurring. Some discussion as to what is causing this trend would be very interesting.

Response: Indeed, especially areas with an increasing LAI_{max} and decreasing woody cover are very interesting and match very well with areas of human induced land conversions. See new Figure 3c and text lines 172-178 and 209-228.

In light of recent research by Reich et al (P. B. Reich, S. E. Hobbie, T. D. Lee, M. A. Pastore, Unexpected reversal of C3 versus C4 grass response to elevated CO₂ during a 20-year field experiment. *Science* 360, 317–320 (2018). doi:10.1126/science.aas9313pmid:2967459) they suggest that we could be expecting to see c4 grass benefiting from elevated co₂. Your results do indicate this trend - it would be of interest to hear how your interpretation of results fits into this debate?

Response: While this field experiment represents a very interesting result in itself, the additional analysis we included in the revised version singling out the areas where positive trends in LAI_{max} dominated over trends of woody cover do not really suggest CO₂ as the likely driver of this phenomenon as these areas of a

difference in trends of higher LAI_{max} than woody cover (Supplementary figure 3) coincide with areas of major human induced land changes from crop expansion into savanna forest ecosystems.

REVIEWERS' COMMENTS:

Reviewer #1 (Remarks to the Author):

Dear authors,

Thank you for a well considered response and careful changes. I have found the changes to be most helpful and from my perspective they have improved the clarity of the manuscript. The manuscript shows an improved consistency in terminology and this improves the interpretation of the results for the reader.

The authors arguments have an improved sense of ecology aboth them, whilst this could be improved a bit, I'm happy to leave it be at this stage.

Looking forward to seeing this work published

Reviewer #2 (Remarks to the Author):

I am grateful for the authors attention to my comments and for their thoughtful responses. I find this manuscript to be of very high quality and the results of broad interest and communicated in a manner that is both clarifying and insightful.